# Aligning Santal Tribe Menu Templates with EAT-Lancet Commission’s Dietary Guidelines for Sustainable and Healthy Diets: A Comparative Analysis

**DOI:** 10.3390/nu16030447

**Published:** 2024-02-02

**Authors:** Sarah Armes, Arundhita Bhanjdeo, Debashis Chakraborty, Harmanpreet Kaur, Sumantra Ray, Nitya Rao

**Affiliations:** 1NNEdPro Global Institute for Food, Nutrition and Health, Cambridge CB4 0WS, UK; s.armes@nnedpro.org.uk (S.A.); d.chakraborty@nnedpro.org.uk (D.C.); h.kaur@nnedpro.org.uk (H.K.); s.ray@nnedpro.org.uk (S.R.); 2Professional Assistance for Development Action (PRADAN), New Delhi 110049, India; b.arundhita@gmail.com; 3School of Biomedical Sciences, Ulster University at Coleraine, Coleraine BT52 1SA, UK; 4Fitzwilliam College, University of Cambridge, Cambridge CB3 0DG, UK; 5School of Global Development, University of East Anglia, Norwich NR4 7TJ, UK

**Keywords:** plant-based diet, indigenous food, cultural diversity, nutrient adequacy, health effects

## Abstract

Background: In the context of global shifts in food systems, this paper explores the unique dietary practices of the Santal tribe, an indigenous group in eastern India, to understand the health, nutrition, and sustainability aspects of their traditional food systems. This study evaluates the nutritional content of the Santal diet in comparison to the EAT-Lancet Commission’s 2019 dietary guidelines for healthy and sustainable diets. Methods: The University of East Anglia, in collaboration with the NNEdPro Global Institute for Food, Nutrition and Health in Cambridge, PRADAN; colleagues in India and local Santal youth, conducted nutritional analyses of traditional Santal recipes. Two menu templates, Kanhu Thali and Jhano Thali, were selected for comparative analysis based on their representation of diverse dietary practices within the Santal community. Nutritional data, including energy as well as the distribution of macronutrients and micronutrients, were compiled and compared with the EAT-Lancet guidelines. Results: The Santal menu templates (nutritionally complete meals) demonstrated alignment with EAT-Lancet recommendations in aspects such as whole grains, starchy vegetables, vegetables, plant-based protein sources, unsaturated fats, and limited added sugars. However, notable deviations included the absence of animal-based protein sources and dairy. The Santal diet showed high protein intake, largely from plant-based sources, and emphasised the importance of whole grains. Seasonal variations in nutritional content were observed between the two templates. Conclusions: While the Santal diet aligns with some aspects of global dietary guidelines, there are notable deviations that underscore the complexity of aligning traditional diets with universal recommendations. The findings emphasise the need for culturally sensitive dietary recommendations that respect traditional diets while promoting sustainability. Research needs to support tailored global guidelines enshrining core principles of nutritional adequacy which are inter-culturally operable in order to accommodate cultural diversity, local practices, and seasonal variations, crucial for fostering sustainable and healthy eating habits in diverse sociodemographic contexts.

## 1. Background

In the last century, global food systems underwent significant changes impacting food supplies, diets, and health outcomes, moving from nutrient deficiency to dietary excess imbalances and now a focus on food system sustainability [1]. Sub-optimal diets, and their downstream metabolic effects, remain a top risk factor for the global burden of disease [2]. An assessment of global dietary quality from 1990 to 2018 found wide differences by nation, age, education, and urbanity [3,4], alongside exerting growing pressure on natural resources, land, biodiversity, and ecosystems [5]. What is clear is a considerable portion of the population, hindered by factors such as poverty, faces barriers to accessing nutritious diets [6].

The environmental impact of dietary choices is significant. It is estimated that, by 2050, diets high in refined sugars, fats, oils, and meats will contribute to an 80% increase in agricultural greenhouse gas emissions and global land clearance [7]. Plant-based diets, which primarily include foods derived from plants like fruits, vegetables, grains, legumes, nuts, and seeds, have the potential to contribute significantly to mitigating climate change. Modelling suggests transitioning towards these diets could lead to a substantial reduction in greenhouse gas emissions, estimated to be between 30% and 55% [5,7]. Operating within Earth’s limits, and managing resources responsibly, necessitates a shift to healthy diets characterised by a diversity of plant-based foods; low amounts of animal source foods; unsaturated rather than saturated fats; and small amounts of refined grains, highly processed foods, and added sugars [8]. Achieving this transformation by 2050 requires substantial dietary alterations and region-specific strategies. Such strategies need to incorporate Double-Duty Actions as set out by the World Health Organisation in 2019 so that we are able to address under- and over-nutrition alongside concurrent micronutrient deficiencies using multifaceted interventions [9].

Indigenous food systems represent complex, culturally embedded approaches to food production and consumption, informed by the traditional knowledge and practices of indigenous communities [10,11]. These systems are characterised by a sustainable, holistic, and locally rooted ethos, incorporating diverse, seasonal, and often wild-harvested food sources. Surrounded by rich agroforestry and diverse ecosystems, indigenous groups possess deep knowledge about their forest environment and its resources. It is acknowledged that traditional and indigenous foods, along with dietary diversity drawn from local ecosystems, serve as robust sources of essential nutrients, contributing significantly to improved health, but also play a pivotal role in mitigating food insecurity [12]. 

India is a host to a diverse range of indigenous communities, also called Scheduled Tribes, making up about 8.6% of the total population [13]. The Santals constitute the largest tribal community in India [14] and have managed to maintain their traditional way of life despite pressures of modernisation and globalisation. Most of the population are cultivators with less than 1 hectare of land [15] and live on very low incomes, often working as day labourers in agriculture [16] and engaging in foraging, hunting, and livestock breeding to supplement their income [17]. One crucial aspect of their diet is the consumption of locally available, indigenous wild foods comprised of culturally important wild plant foliage, fungi species, vegetables, fruits, locally raised livestock, and small aquatic species [17,18]. However, there has been a decline in consumption of traditional foods in these communities [19], replaced by more convenient, packaged foods from the market. The high prevalence of malnutrition, anaemia, and chronic energy deficiency in the Santal community residing in different states of India [20] can be partly attributed to this shift.

In addition to their foraging practices, the Santal tribe engage in small-scale agriculture and maintain kitchen gardens. These cultivated spaces yield a variety of green leafy vegetables, which are integral components of their daily dietary intake [20]. What distinguishes the dietary practices of indigenous communities, such as the Santal tribe, is their cultivation and consumption of native crops, known for their higher levels of essential micronutrients compared to non-native plant varieties [21]. Emphasising indigenous crops is crucial for ensuring households have access to more diverse and nutritious diets. Research has demonstrated that even limited consumption of indigenous food varieties substantially improves intake of specific micronutrients, including riboflavin, iron, vitamin A, calcium, zinc, and folic acid [17]. Additionally, several indigenous foods have been shown to contain high levels of bioactive components with anti-inflammatory properties amongst other potential health benefits. This underscores the importance of preserving and promoting diverse and indigenous food sources among the Santal tribe to enhance overall well-being and nutrition. Research often overlooks the evaluation of how local diets and cultural preferences intersect global guidelines, such as the EAT-Lancet guidelines, leaving a gap in our understanding of the practical implementation of these global dietary recommendations in diverse cultural contexts.

The EAT-Lancet Commission’s 2019 dietary guidelines promote a “Planetary Health Diet” that emphasises plant-based foods, reduces animal products, and prioritises sustainability [8]. Key messages highlight the critical importance of transforming our food systems to address both health and environmental challenges by offering a globally adaptable approach for individuals, governments, and healthcare professionals to improve personal health while reducing the environmental impact of food production [8]. While the EAT-Lancet Commission has established global objectives for promoting healthy and sustainable diets, implementing these guidelines in culturally diverse and region-specific contexts presents significant challenges. Recognising the importance of diverse food cultures and aligning them with EAT-Lancet benchmarks represents a crucial endeavour to enhance their acceptance and effectiveness, ensuring sustainable and healthy dietary choices are accessible and relevant to people from all backgrounds. 

This study aims to assess the alignment between Santal menu templates, derived from indigenous recipes, and the dietary guidelines proposed by the EAT-Lancet Commission for healthy and sustainable diets. 

## 2. Methods

The University of East Anglia (UEA), in collaboration with NNEdPro Global Institute for Food, Nutrition and Health in Cambridge, PRADAN; colleagues in India; and local Santal youth, investigated the nutritional adequacy of traditional recipes consumed by the Santal tribe in eastern India and provided suggested alterations to address nutritional deficiencies and support the dietary diversity, practical application, and sustainability of food systems [22]. From our prior research, we have successfully created a Santal cookbook, with a total of nine daily menu templates, putting together the recommended combination of indigenous recipes with nutritive values so as to address poor micro- and macronutrient intakes prevalent in the Santal community. The nine menu templates are seasonally categorised based on the availability of specific food items.

### 2.1. Menu Template Selection

For the comparative analysis with the EAT-Lancet guidelines, two out of the nine Santal menu templates from the cookbook were chosen (Kanhu Thali and Jhano Thali). Thali is a local and Hindi term for plate. In this context, the term “plate” or “template” refers to a representation of three recommended meals to be consumed in a single day. The process of selecting menu templates aimed to encompass a range of meal structures and ingredient compositions that represent the diverse dietary practices within the Santal community. Additionally, careful consideration was given to seasonality, with menu templates selected to cover both summer and winter seasons, thus accounting for dietary variations.

### 2.2. Kanhu Thali (Winter Season—November to February)

This menu template is designed to cater to dietary preferences of the Santal community during the winter season (Figure 1). It consists of a variety of meals throughout the day, ensuring a comprehensive representation of daily dietary patterns. The breakdown of meals includes the following:

Meal 1 (Morning): crushed sweet corn boiled with horsegram (*Jonra Dakaa* and *Kurthi Daal*).

Meal 2 (Day): rice, flat beans, wild leafy vegetables, and dried fish (*Malhan Daal Ohoy Ara* and *Sukhi Machli*).

Meal 3 (Evening): wheat flour chapatis, black-eyed beans, and drumstick leaves (*Lupung Ara Peetha*, *Ghanghra Daal* with *Lal Ara*, and *Munga Ara*).

### 2.3. Jhano Thali (Late Summer to Monsoon)

This menu template caters to dietary preferences of the Santal community during the transition from late summer to the monsoon season (Figure 2). It offers a glimpse into the diversity of meals consumed during this period, reflecting the seasonal availability of ingredients. The breakdown of meals includes the following:

Meal 1 (Morning): rice, sweet potato leaves, and black-eyed beans (*Sakarkand*/*Alu Ara* and *Ghanghra Daal*)

Meal 2 (Day): mahua flower (*Madhucaa longifolia*) with sesame seeds (*Matkom Tilmin Lathe*)

Meal 3 (Evening): rice, wild mushroom curry, chicken egg curry, and mango pickle (*Mocha Oo Uttu*, *Sim Bili Uttu* and *Ool Ka Achar*)

### 2.4. Compilation of Nutritional Data

The nutritional information for each menu template was analysed using Nutritics Professional Premium software (Research Edition, v5.64, Dublin, Ireland, Nutritics, 2022, Libro). For food ingredients that were specific to the Santal tribe, and which were not available through Nutritics, advice was provided by the NNEdPro India team based on nutritive values of Indian foods from the National Institute of Nutrition (NIN). Where the nutritive values were not available through either Nutritics or NIN the next closest food item was used. Each template was screened for total energy, protein, carbohydrates, fibre, sugars, saturated, monounsaturated and polyunsaturated fat, omega-3, trans-fatty acids, sodium, potassium, calcium, magnesium, iron, zinc, copper, manganese, selenium, iodine, vitamin A, vitamin D, vitamin E, vitamin K1, thiamine, riboflavin, pantothenic acid, folates, vitamin B6, Vitamin B12, and vitamin C. Descriptive statistics, including means and standard deviations, were calculated for relevant nutritional parameters.

### 2.5. Comparison with EAT-Lancet Guidelines

The nutritional contents of each menu template were evaluated regarding their alignment with the EAT-Lancet healthy reference diet [8]. This analysis aimed to uncover areas of convergence or divergence, with a specific emphasis on micronutrient intake measured in grams and total energy expressed in calories per day. 

## 3. Results

Table 1 presents the dishes and ingredients for two Santal menu templates: Jhano Thali and Kanhu Thali. Both menu templates feature dishes that incorporate locally sourced seasonal ingredients, showcasing the Santal tribe’s adaptability to their environment and their ability to incorporate foraged items into their culinary traditions.

Table 2 highlights the nutritional differences between Kanhu Thali and Jhano Thali menu templates, highlighting seasonal variations in nutritional content. In terms of macronutrients, Kanhu Thali provides 2289 kcal of energy, while Jhano Thali provides 3427 kcal, averaging 2858 kcal. The carbohydrate content varies between menu templates, with Jhano Thali providing 524 g compared to Kanhu Thali, which provides 332.4 g, together averaging 428.3 g. Conversely, Kanhu Thali offers more protein with 106.1 g compared to Jhano Thali (90.3 g), together averaging 98.2 g. In terms of total fat, Jhano Thali provides 106.1 g of fat, while Kanhu Thali provides 59.7 g, together averaging 82.9 g. Much of this is from mono and polyunsaturated fat, 47.0 g and 38.0 g in Jhano Thali and 29.8 g and 15.4 g in Kanhu Thali, respectively. Additionally, Kanhu Thali provides higher fibre content with an average of 37.1 g, compared to 35.3 g for Jhano Thali, with an average of 36.2 g across the two templates. On average, carbohydrates contribute 59.9%, proteins contribute 13.7%, and fats contribute 26.1% to total energy intake.

Regarding micronutrients, Kanhu Thali exhibited higher levels of sodium (10,789 mg), potassium (4968 mg), calcium (1420 mg), iron (44.5 mg), vitamin A (2848 μg), vitamin K1 (42.1 μg), riboflavin (1.9 mg), niacin (42.8 mg), vitamin B6 (3.5 mg), folate (1262.6 μg), vitamin B12 (2.1 μg), and vitamin C (520 mg) in comparison to Jhano Thali (sodium: 3971 mg; potassium: 3606 mg; calcium: 1126 mg; iron: 39.5 mg; vitamin A: 1972 μg; vitamin K1: 30.8 μg; riboflavin: 0.9 mg; niacin: 39.5 mg; vitamin B6: 2.0 mg; folate: 620.9 μg; vitamin B12: 1 μg,; and vitamin C: 226 mg). Conversely, Jhano Thali displayed greater quantities of zinc (16.0 mg), copper (4.0 mg), manganese (6.5 mg), selenium (117.2 μg), iodine (38.8 μg), vitamin D (1.6 μg), vitamin E (22.1 mg), and thiamin (2.6 mg) compared to Kanhu Thali (zinc: 11.3 mg; copper: 2.4 mg; manganese: 6.1 mg; selenium: 80.7 μg; iodine: 12.4 μg; vitamin D: 0.0 μg; vitamin E: 9.6 mg; and thiamin: 2.2 mg).

Table 3 compares the food weight (g/day) and caloric intake (kcal/day) of two Santal menu templates with the EAT-Lancet recommendations based on the different food groups. Additionally, it highlights the total macronutrients (g/day) of Kanhu Thali and Jhano Thali.

The average of the two templates generally aligns with the EAT-Lancet dietary guidelines, exceeding the recommended values for whole grains, starchy vegetables, and vegetables, while meeting recommendations for protein sources (including eggs, fish, and legumes), added fats (unsaturated oils), and added sugars. In terms of whole grains, Kanhu Thali contains 450.1 g of whole grains, providing 1103 kcal and 269.5 g of macronutrients. Jhano Thali, on the other hand, offers 500 g, providing 1706 kcal and 423.2 g of macronutrients. The average food weight across both templates is 475 g, providing 1404.5 kcal and 346.35 g of total macronutrients (of which 306 g are from carbohydrates). Although this exceeds the EAT-Lancet recommendations of 232 g for whole grains, it falls within the recommended range of 0–60% of daily energy intake. Regarding starchy vegetables, the average food weight was 150 g, providing 150.5 kcal and 36.8 g of macronutrients (of which 33.9 g are from carbohydrates), which exceeds the EAT-Lancet recommendation of 0–100 g. In the case of overall vegetable consumption, Kanhu Thali contains 623.2 g and Jhano Thali 284.0 g, together with an average of 453.6 g. This meets the EAT-Lancet recommendations of between 200–600 g. In regard to protein sources, the Santal menu templates provide an average of 25 g of eggs and 35 g of fish (providing 12.1 g and 3.5 g of protein, respectively), which is within the EAT-Lancet recommendations. In the legumes category, the average intake is 100 g, providing 59 g of total macronutrients (an average of 38.2 g of carbohydrates and 19.2 g of protein). When it comes to added fats, the average across both menu templates is 37.7 g (of which 37.5 g are from fats), aligning with the EAT-Lancet recommendation of 20–80 g. Additionally, in terms of added sugars, both Kanhu Thali and Jhano Thali have no recorded intake, which is within the EAT-Lancet recommendations of 0–31 g.

However, the Santal menu templates did not meet recommendations for fruit and dairy products. The average total fruit content of both menu templates is 97 g and 39.4 kcal, which falls slightly below the EAT-Lancet recommendation of 100–300 g for a daily intake of 126 kcal. Culturally, the Santal community, as well as several other indigenous groups of eastern India, do not consume dairy products, in contrast to the EAT-Lancet recommendation of 250 g.

## 4. Discussion

The comparative analysis presented in this study sheds light on the unique dietary practices of the Santal tribe and offers valuable insights into how to tailor global dietary recommendations to local cultures, fostering a culturally sensitive approach to promoting sustainable and healthy eating habits. Our analysis highlights several aspects of the Santal menu templates are in accordance with the EAT-Lancet recommendations [8], including an emphasis on whole grains, the incorporation of starchy vegetables and other vegetables, the inclusion of plant-based protein sources such as legumes, a focus on unsaturated fats, the limited consumption of added sugars, and the use of locally sourced ingredients. 

The central role of whole grains in the Santal diet is evident in the high intake of rice, wheat, and corn. Whole grains not only provide essential nutrients but contribute to dietary fibre, thereby promoting digestive health [23]. Rice is a primary component of main meals in the Santal tribe. This is consistent with other studies which show the daily diet of the Santal community contains watery or hot rice with leafy vegetables, pulses or small amounts of meat or fish [24]. Our findings indicate that the Santal templates provide an average of 475.1 g of whole grains, which is above both the EAT-Lancet recommendations of 232 g and the Recommended Dietary Allowances (RDA) for Indians of 400 g, as outlined in ‘Nutritive Value of Indian Food’ published by NIN in Hyderabad [25]. The substantial intake of whole grains in these templates provides a significant portion of dietary energy, thereby contributing to daily calorie intake. The two menu templates provided an average of 428.3 g of carbohydrates, equivalent to 1713.2 kcal or approximately 60% of total energy intake. This aligns with the Institute of Medicine’s recommended macronutrient distribution range for carbohydrates (45–65% of total energy) [26]. The relatively high contribution of carbohydrates to total energy requirements is important when considering the active lifestyle of these tribal communities, involving moderate (62.8%) to heavy (23.0%) physical activities like grazing, harvesting, and ploughing [27]. Furthermore, Kanhu Thali notably provides excessive amounts of salt (10,789 mg), which raises concerns about the potential impact on cardiovascular health. However, due to high levels of physical activity among Santal adults, they may require higher amounts of sodium due to sodium sweat loss [28]. 

While the Santal templates provide an average of 98.2 g of protein (13.7% of energy contributions), it is important to consider the protein quality. The high levels of rice consumption may lead to a high aggregated protein intake but of poor quality, with a suboptimal amino acid profile and low protein digestibility [29]. Another important source of protein in the Santal tribe is legumes and pulses, of which the average between the two templates is 100 g (242.75 kcal and 59 g total macronutrients), which meets the EAT-Lancet recommendations of 50 g as well as the RDA for Indians of 80 g. Legumes, unlike perishable animal protein and milk, serve as a valuable protein source for the Santals [29], as they offer a non-perishable protein option that can be purchased from external markets and stored for extended periods. However, the presence of phytates in the Santal diet should be noted, driven by the relatively high consumption of whole grains, legumes, and specific vegetables. 

One of the notable deviations from the EAT-Lancet recommendations is the absence of animal-based protein sources, including lamb, beef, pork, and poultry, as well as dairy products. Consumption of high-quality protein, such as meat, poultry, and milk products, was low, as reported in other Indian tribal populations [30,31]. However, the Santal community consume protein-rich food items accessed from ponds including indigenous fish and snails [20]. This highlights the adaptability of the Santal tribe in sourcing their protein requirements from local, culturally accepted, and readily available sources. Additionally, the absence of dairy in these menu templates also deviates from the RDA for Indians, which recommends 300 g of milk consumption per day. This prompts a discussion around the reasons behind this omission and the role of cultural practices, alongside the limited availability of dairy in the region. 

Additionally, our study highlights that on average the Santal templates contained 453.6 g of vegetables and 97 g of fruit. This exceeds both the EAT-Lancet recommendations and the Indian RDA of 300 g for vegetables and falls slightly below the EAT-Lancet recommendations of 200 g (100–300 g) for fruit. However, our findings are in contrast with the findings from the National Nutrition Monitoring Bureau (NNMB) surveys, which found a vegetable consumption of 43 g per day [32]. Santals consume a variety of locally available seasonal indigenous fruits typically influenced by their geographical proximity to forest areas. However, another study reported that the frequency of fruit consumption is low in the santal community [20]. Moreover, it is important to note that certain traditional and culturally significant food items including mahua flower, sesame seeds, and spices such as cumin and turmeric powder lack a specific category in the EAT-Lancet guidelines, yet they hold considerable importance in the Santal dietary context. This highlights the complexity of aligning traditional diets with universal recommendations. It also underscores the need for cultural sensitivity and the recognition that traditional diets may meet nutritional needs in ways that differ from conventional dietary norms. 

Whilst the current study looks at the alignment of only two menu templates, it is important to consider that the Santal diet includes a variety of other indigenous foods including green leafy vegetables, meat and meat products, fruits, vegetables, and roots and tubers during specific seasons or throughout the year depending upon availability [20]. Additional menu templates from the Santal cookbook feature seasonally categorised indigenous foods including several green leafy vegetables and wild mushrooms; fruits such as mango, ber (Ziziphus Mauritiana), and mahua; vegetables such as radish, jackfruit, and okra; and flesh foods such as prawns, dry fish, shellfish, crabs, red ants, and pork. Sugar is consumed once or twice weekly [20]. 

In regard to micronutrients, previous studies on the Santal tribe have highlighted substantial micronutrient deficiencies, particularly in calcium, iron, vitamin B2, folate, and vitamin B12 [20,33]. However, nutritional analysis reveals that Kanhu Thali and Jhano Thali menu templates align with recommended levels of these essential nutrients. Recommendations in menu templates include the consumption of green leafy vegetables and mahua flower, which are sources of folate, calcium, and non-heme iron, as well as greater levels of eggs and fish, which contain vitamin B12, zinc, and heme iron. However, we found deficiencies in key micronutrients, including iodine (25.6 μg), vitamin D (0.8 μg), and vitamin K1 (36.5 μg), all of which fall below recommended levels by 82.9%, 92.0%, and 59.4%, respectively. Iodine deficiency is a prominent health problem in India, with surveys highlighting that no state is free from iodine deficiency disorder [34]. The menu templates aimed to increase the consumption of iodine-rich foods such as egg and fish; however, average iodine content remained below recommendations. Furthermore, despite falling short of recommended vitamin D levels in the menu templates, a study on Santal women found all women had sufficient vitamin D levels [33]. This could be at-tributed to their lifestyle, predominantly involving agricultural and foraging activities, suggesting they may meet their vitamin D requirements through sunlight exposure rather than dietary intake. Our analysis also revealed the Santal menu templates were deficient in vitamin K1 by 59.4%. Typically, green leafy vegetables constitute the primary source of vitamin K1, contributing to approximately 60% of total intake [35]. To address these deficiencies, there is a critical need to educate tribal populations on locally available and culturally accepted food sources rich in these nutrients, as well as supplementation, fortification, and suitable agricultural practices.

In more recent years, dietary patterns within the Santal community are undergoing transformative shifts to energy-rich foods [36]. This transformation, influenced by various factors embedded in the socio-cultural and policy environment of India, has led to underutilisation of locally available and indigenous food sources [17]. The evolving dietary landscape is shaped by the complex interplay between education and awareness campaigns, economic changes, generational transitions, and shifts in agricultural practices, which have a profound impact on dietary choices. For instance, a study conducted in the Santal community of Jharkhand, India highlighted a shift in the consumption of traditional rice varieties (with higher fibre and nutrient content) towards polished white rice [37]. This transition may lead to a decline in the nutritional quality of rice consumed, emphasising the importance of preserving traditional crop diversity. Acknowledging and safeguarding the traditional ecological knowledge of indigenous communities has the potential to bring about positive impacts in key sectors like agriculture, forest and biodiversity management, infrastructure, and fishing, ultimately contributing to addressing the broader challenges of climate change and sustainability [38]. This underscores the need for tailored public health interventions that consider specific socio-economic and generational contexts. However, translating global dietary guidelines into local realities presents a set of unique challenges. The accessibility and availability of foods can act as barriers; the Santal community, like many indigenous communities, faces challenges in accessing certain foods that align with the EAT-Lancet guidelines. Furthermore, the preservation of cultural culinary traditions must be balanced with recommended dietary changes. Research in this domain should explore cultural implications of dietary shifts, particularly in heritage preservation and intergenerational knowledge transfer. At the same time, cultural sensitivity in health campaigns and other policy interventions also warrants thorough investigation to ensure they align with local resources and culture. Additionally, cross-cultural comparisons can provide valuable insights into dietary patterns across diverse cultural settings, enabling the identification of commonalities and distinctions in dietary choices, and the opportunities and challenges they bring in the process of moving towards planetary health diets.

## 5. Conclusions

The findings from the Santal tribe’s menu templates highlight the importance of tailoring dietary guidelines to accommodate cultural diversity, local practices, and seasonal variations. Recognising and respecting traditional diets and their adaptability to local environments is key to promoting sustainable and healthy eating. While global dietary recommendations play a vital role in public health, they should be crafted in a way that allows for cultural sensitivity and regional adaptability. It is essential to undertake action-orientated research to tailor global guidelines to enshrine core principles of nutritional adequacy that are inter-culturally operable and able to accommodate cultural diversity, local practices, and seasonal variations. This is crucial for fostering sustainable and healthy eating habits in diverse sociodemographic contexts.

## Figures and Tables

**Figure 1 nutrients-16-00447-f001:**
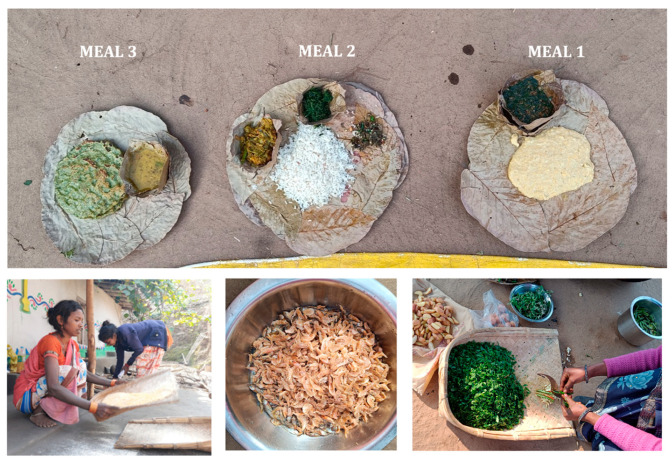
Top: Kanhu Thali. Bottom: preparation of Kanhu Thali—woman sieving the corn (**left**), dried fish (**centre**), and woman cleaning and cutting the wild green leafy vegetables (**right**).

**Figure 2 nutrients-16-00447-f002:**
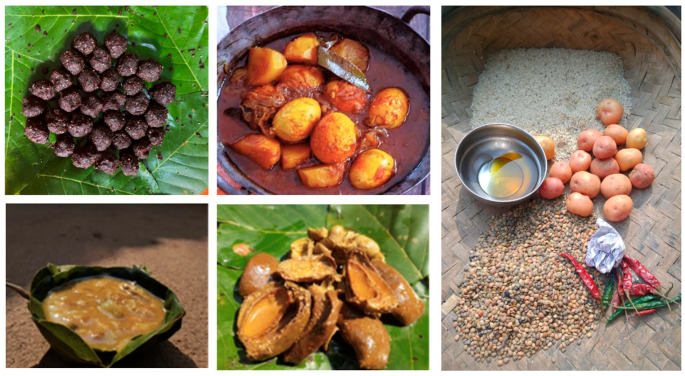
Preparation of Jhano Thali ((**top left**): mahua flowers with sesame seeds; (**top middle**): chicken egg curry; (**bottom left**): Black eyed beans; (**bottom middle**): mango pickle; (**right**): rice, horsegram, and other raw materials for meal preparation).

**Table 1 nutrients-16-00447-t001:** List of dishes and ingredients for Kanhu Thali and Jhano Thali menu templates.

Menu Template	Dishes	Ingredients
Kanhu Thali (Winter season—November to February)	Sweet corn *(Jonra Dakaa*)	Dried, slightly smashed sweet corn 150 g and salt 3 g
	Horsegram (*Kurthi Daal*)	Salt 1 g; oil 5 g; red chilli 20 g; cumin 2 g; tomatoes 100 g; potatoes 100 g; aubergine 150 g; and dried, semi-thrashed lentils (brown) 30 g
	Rice	Parboiled rice/fermented rice 250 g
	Flat beans (*Malhan Daal*)	Flat bean seeds (malhan) 50 g, cumin 2 g, oil 2 g, red chilli 1 g, salt 2 g, chilli green 3 g, and turmeric powder 1 g
	Wild green leaves *(Ohoy Ara*)	Wild leafy vegetables 200 g, oil 5 g, salt 2 g, garlic 2 g, and chilli green 3 g
	Dried fish *(Sukhi Machli*)	Whitefish 70 g, onions 10 g, oil 4.6 g, red chilli 5 g, garlic 5 g, and ginger 4 g
	Wheat flour chapati *(Lupung Ara Peetha*)	Flour wheat wholemeal 50 g, chilli 3 g, cumin 5 g, garlic 2 g, salt 2 g, oil 13.8 g, and uncultivated green leafy vegetables 50 g
Jhano Thali (late summer to monsoon seasons—June to September)	Sweet potato leaves and black-eyed beans (*Sakarkand*/*Alu Ara*)	Sweet potato leaves 100 g, salt 2 g, oil 5 g, red chilli 5 g, garlic 3 g, and sweet potato 200 g
	Black-eyed beans *(Ghanghra Daal*)	Black-eyed beans 60 g, cumin 2 g, oil 2 g, red chilli 20 g, salt 2 g, and chilli green 3 g
	Rice	Parboiled rice/fermented rice 250 g
	Mahua flowers with sesame seeds (*Matkom Tilmin Lathe*)	Semi-dried mahua flower 50 g and sesame seeds 100 g
	Wild mushroom curry *(Mocha Oo Uttu*)	Oil 5 g, mushrooms 100 g, salt 1 g, ginger 3 g, garlic 2 g, onions 10 g, and chilli green 5 g
	Egg curry (*Sim Bili Uttu*)	Oil 5 g, cumin 2 g, salt 1 g, chilli green 3 g, garlic 3 g, ginger 3 g, onions 10 g, chicken eggs 50 g, and turmeric powder 2.2 g
	Mango pickle *(Ool Ka Achar*)	Salt 3 g, turmeric powder 1 g, oil 10 g, and green mango 100 g

**Table 2 nutrients-16-00447-t002:** Nutritional content of Kanhu Thali and Jhano Thali menu templates.

Nutrient	Kanhu Thali (Winter Season)	Jhano Thali (Late Summer to Monsoon)	Average of the Two Templates (Mean (SD))
Energy (Kcal)	2289.0	3427.0	2858.0 (804.7)
Contribution of Carbohydrate to Total Energy (%)	58.1	61.2	59.9 (2.2)
Carbohydrate (g)	332.4	524.1	428.3 (135.6)
Contribution of Protein to Total Energy (%)	18.5	10.5	13.7 (5.7)
Protein (g)	106.1	90.3	98.2 (11.2)
Contribution of Fat to Total Energy (%)	23.5	27.9	26.1 (3.1)
Fat (g)	59.7	106.1	82.9 (32.8)
Starch (g)	297.2	480.6	388.9 (129.7)
Fibre (g)	37.1	35.3	36.2 (1.3)
Sugars (g)	18.5	41.2	29.9 (16.1)
Saturated Fat (g)	5.9	16.5	11.2 (7.5)
Monounsaturated fat (g)	29.8	47.0	38.4 (12.2)
Polyunsaturated fat (g)	15.4	38.0	26.7 (16.0)
Omega 3 (*n*-3) (g)	3.5	3.3	3.4 (0.1)
Trans-fatty acids (g)	0.0	0.0	0.0 (0.0)
Sodium (mg)	10,789.0	3971.0	7380.0 (4821.1)
Potassium (mg)	4968.0	3606.0	4287.0 (963.1)
Calcium (mg)	1420.7	1126.7	1273.7 (207.9)
Magnesium (mg)	559.5	643.6	601.6 (59.5)
Iron (mg)	44.5	39.5	42.0 (3.5)
Zinc (mg)	11.3	16.0	13.7 (3.3)
Copper (mg)	2.4	4.0	3.2 (1.1)
Manganese (mg)	6.1	6.5	6.3 (0.3)
Selenium (μg)	80.7	117.2	99.0 (25.8)
Iodine (μg)	12.4	38.8	25.6 (18.7)
Vitamin A (μg)	2848.0	1972.0	2410.0 (619.4)
Vitamin D (μg)	0.0	1.6	0.8 (1.1)
Vitamin E (mg)	9.6	22.1	15.9 (8.8)
Vitamin K1 (μg)	42.1	30.8	36.5 (8.0)
Thiamin (B1) (mg)	2.2	2.6	2.4 (0.3)
Riboflavin (B2) (mg)	1.9	0.9	1.4 (0.7)
Niacin (B3) (mg)	42.8	39.5	41.2 (2.3)
Pantothenic Acid (B5) (mg)	7.1	9.2	8.2 (1.5)
Vitamin B6 (mg)	3.5	2.0	2.8 (1.1)
Folates (B9) (μg)	1262.6	620.9	941.8 (453.8)
Vitamin B12 (μg)	2.1	1.0	1.6 (0.8)
Vitamin C (mg)	520.0	226.0	373.0 (207.9)

**Table 3 nutrients-16-00447-t003:** Food weight, total macronutrients, and caloric intake of two Santal menu templates with the EAT-Lancet healthy reference diet.

	Kanhu Thali			Jhano Thali			Average			EAT-Lancet Recommendations	
	Food Weight (g/day)	Total Macronutrients (g/day)	Energy in Calories (kcal/day)	Food Weight (g/day)	Total Macronutrients (g/day)	Energy in Calories (kcal/day)	Food Weight (g/day)	Total Macronutrients (g/day)	Energy in Calories (kcal/day)	Food Weight (g/day)	Energy in Calories (kcal/day)
Whole grains											
Rice, wheat, corn, and other	450.1	269.5	1103	500.0	423.2	1706	475.05	346.4	1404.5	232 (0–60% of energy)	811
Tuber or starchy vegetables											
Potatoes and cassava	100.0	16.5	67	200.0	57	234	150	36.8	150.5	50 (0–100)	39
Vegetables											
All vegetables	623.2	55	238.7	284.0	49.8	220.7	453.6	52.4	229.7	300 (200–600)	78
Dark green vegetables	350.1	42.5	185.4	150.0	43.9	196	250.05	43.2	190.7	100	23
Red and orange vegetables	100.0	3.6	14.9	0	0	0	50	1.8	7.45	100	30
Other vegetables	173.1	8.9	38.4	134.0	5.9	24.7	153.55	7.4	31.6	100	25
Fruits											
All fruit	58.0	4.9	23	136.0	13.4	55.8	97	9.15	39.4	200 (100–300)	126
Dairy foods											
Whole milk or derivative equivalents (e.g., cheese)	0	0	0	0	0	0	0	0	0	250 (0–500)	153
Protein sources											
Beef and lamb	0	0	0	0	0	0	0	0	0	7 (0–14)	15
Pork	0	0	0	0	0	0	0	0	0	7 (0–14)	15
Chicken and other poultry	0	0	0	0	0	0	0	0	0	29 (0–58)	62
Eggs	0	0	0	49.9	11.8	71	24.95	5.9	35.5	13 (0–25)	19
Fish	70.3	25	104	0	0	0	35.15	12.5	52	28 (0–100)	40
Legumes											
Dry beans, lentils, andpeas	140	73	301.5	60.0	44.9	184	100	59	242.8	50 (0–100)	172
Soy foods	0	0	0	0	0	0	0	0	0	25 (0–50)	112
Peanuts	0	0	0	0	0	0	0	0	0	25 (0–75)	142
Tree nuts	0	0	0	100.0	77	613	50	38.5	306.5	25	149
Added fats											
Palm oil	0	0	0	0	0	0	0	0	0	6.8 (0–6.8)	60
Unsaturated oils	40.4	40.1	360.6	35.0	34.9	313	37.7	37.5	336.8	40 (20–80)	354
Dairy fats (included in milk)	0	0	0	0	0	0	0	0	0	0	0
Lard or tallow	0	0	0	0	0	0	0	0	0	5 (0–5)	36
Added sugars											
All sweeteners	0	0	0	0	0	0	0	0	0	31 (0–31)	120

## Data Availability

Dataset available on request from the authors.

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
