# Peer review of "Aligning Santal Tribe Menu Templates with EAT-Lancet Commission’s Dietary Guidelines for Sustainable and Healthy Diets: A Comparative Analysis"

_nutrients, 2024, doi:10.3390/nu16030447_

Round 1

Reviewer 1 Report

Comments and Suggestions for Authors

Interesting paper, on a very important topic, thought from a different perspective.   

The sentence in lines 47 to 5 needs to be revised. Poor English. 

Attention to how the words break across lines (e.g. responsibly)

Line 123 - methods - it is not clear whether the cookbook is a result of the whole research, or if the cookbook boo result from preliminary research to select the menus to be evaluated. 

Line 131 - sentence not clear

Table 3 - Energetic intake (instead of Caloric intake) - we can measure energy in Calories or Kjoules. 

I believe the discussion should include a broader view of the other foods from the Santal menus vs EAT lancet diet. There is no mention of fruit, vegetables or sugars.  

Comments on the Quality of English Language

Apart from one sentence, English is good. 

Author Response

Reviewer 1:

Comment:

The sentence in lines 47 to 5 needs to be revised. Poor English. 

Response:

We have revised the sentence structure, and spilt this into two separate sentences to make our point clearer to the reader.

Comment:

Attention to how the words break across lines (e.g. responsibly)

Response:

We appreciate your keen observation regarding word breaks. While we have considered this point, it's important to note that the final decisions on formatting, including word breaks, rest with the editors responsible for refining the paper's layout and presentation.

Comment:

Line 123 - methods - it is not clear whether the cookbook is a result of the whole research, or if the cookbook boo result from preliminary research to select the menus to be evaluated. 

Response:

We have amended the beginning of this sentence to make it clearer that the initial research and development of the Santhal cookbook was completed before drafting this paper.

Comment:

Line 131 - sentence not clear. ‘Here, plate or template connotes three recommended meals in one day.’

Response:

Again, this was revised to make it clear to the reader what we refer to as the ‘templates’.

Comment:

Table 3 - Energetic intake (instead of Caloric intake) - we can measure energy in Calories or Kjoules. 

Response:

In Table 3, we have amended the column heading to ‘Energy in Calories (kcal/day)’

Comment:

I believe the discussion should include a broader view of the other foods from the Santal menus vs EAT lancet diet. There is no mention of fruit, vegetables or sugars.  

Response:

Thank you for this comment. We have included some discussion points around other food sources found in the Santal cookbook, and given specific examples to fruits, vegetables and sugar intake, with reference to previous studies and recommendations and comparing them to our findings.

Reviewer 2 Report

Comments and Suggestions for Authors

interesting manuscript titled "Aligning Santal Tribe Menu Templates with EAT-Lancet Com-2 mission's Dietary Guidelines for Sustainable and Healthy Di-3 ets: A Comparative Analysis"

I have only 2 comments that may clarify some doubts for readers:

1. Table 3 indicates: Macronutrient intake, does it refer to grams of foods or preparations? I think it would be better as grams of foods.

2. In the same table, a low consumption of fruits is shown, this idea is not developed in the discussion, I find it interesting that a tribe culturally does not like fruits (I don't think they don't have them in their environment), and I see that they reach the recommendation of 5 a day (400 g/day) basically by consuming vegetables, please clarify

Author Response

Reviewer 2:

Comment:

Table 3 indicates: Macronutrient intake, does it refer to grams of foods or preparations? I think it would be better as grams of foods.

Response:

Yes, this is written as grams per day (g/day).

Comment:

In the same table, a low consumption of fruits is shown, this idea is not developed in the discussion, I find it interesting that a tribe culturally does not like fruits (I don't think they don't have them in their environment), and I see that they reach the recommendation of 5 a day (400 g/day) basically by consuming vegetables, please clarify.

Response:

Thank you for this comment, and we have made sure to include a section on fruit and vegetable consumption and availability in the discussion.

Reviewer 3 Report

Comments and Suggestions for Authors

The authors present an assessment of the compliance of two traditional Indian food patterns with the Lancet EAT program.

I read the paper with great interest, as it is intriguingly specific and "out-of-the-box" for a large group of potential readers. Thanks for this inspiration and for widening the scope of your global audience.

The overall rationale of the study is fine.

Introduction: Fine.

Methods: Both food patterns are described as typical for a certain time period. However, those time periods together only cover part of a full year. What about the other months?

Results: Average food composition (averaged over the two patterns) seems to reflect the different duration of the time periods, being typical for both patterns, as it is not the simple mean of both composition data (e.g: "Kanhu Thali provides 2290.0 kcal of energy, while Jhano Thali provides 3428 kcal, averaging 2775 kcal." - But the average of 2290 and 3428 kcal would be 2859 kcal). Please clarify.

Table 1 does not clearly show, which foods belong to the first and second pattern. A horizontal split line is necessary.

Please use adequate decimals, reflecting plausible raw data precision and being consistent for both diet patterns.

Please standardize nutrient composition to the same amount of calories, e.g. 2700 kcal/d for a physically active male adult, or whatever amount of calories is suitable for comparison to the EAT pattern and representative for a typical adult person of that geographical region.

Please add relative contribution of macronutrient calories (%kcal) to table 2.

Please add data for magnesium, copper, chrome, molybdene, manganese, Vitamin K, biotin.

Please add data for uric acid/purines, phytate and other undesired natural food components.

Please give an estimation of the PAL for people in the respective regions. High intake of digestible carbohydrates is acceptable, where physical activity allows for compensation.

Table 3: The table head "macronutrient intake" seems misleading; as the EAT recommendations are certainly not about total macronutrients of specific food groups rather than carbs/fat/protein separately or total amount of specific food groups, this should be amended.

Discussion: Both patterns lack iodine, B12; maybe more (see missing micronutrients listed above). Khanu Thali provides excessive amounts of salt. Excessive amounts of carbohydrates may be detrimental for patients with risk for / diagnosis of type 2 diabetes or obesity. Please address these topics.

Is there epidemiological data about the frequency and magnitude of malnourishment or specific nutrient deficiency in people traditionally following these food patterns?

Author Response

Reviewer 3 (Round 1):

Comment:

Methods: Both food patterns are described as typical for a certain time period. However, those time periods together only cover part of a full year. What about the other months?

Response:

Thank you for your comment. While we delved into detail for only two of the nine menu templates, one for summer and the other for winter, we've sought to acknowledge seasonal variability in specific ingredients in the discussion section. Additional menu templates from the Santal Cookbook feature seasonally categorized indigenous foods, encompassing green leafy vegetables, fruits like mango, ber (Ziziphus Mauritiana), and mahua, as well as vegetables such as radish, jackfruit, and okra. The array includes flesh foods like prawns, dry fish, shellfish, crabs, red ants, and pork, along with wild mushrooms.

Comment:

Results: Average food composition (averaged over the two patterns) seems to reflect the different duration of the time periods, being typical for both patterns, as it is not the simple mean of both composition data (e.g: "Kanhu Thali provides 2290.0 kcal of energy, while Jhano Thali provides 3428 kcal, averaging 2775 kcal." - But the average of 2290 and 3428 kcal would be 2859 kcal). Please clarify.

Response:

Thank you for pointing this out, the means and SD have been corrected for table 2.

Comment:

Table 1 does not clearly show, which foods belong to the first and second pattern. A horizontal split line is necessary.

Response:

We have added a seperation line between the two menu templates to make it clearer to the reader which foods belong to each template.

Comment:

Please use adequate decimals, reflecting plausible raw data precision and being consistent for both diet patterns.

Response:

We have made sure to consistently use 1 decimal place in the tables.

Comment:

Please standardize nutrient composition to the same amount of calories, e.g. 2700 kcal/d for a physically active male adult, or whatever amount of calories is suitable for comparison to the EAT pattern and representative for a typical adult person of that geographical region.

Response:

We opted to present unstandardised values in the methods section to provide a transparent and detailed account of the specific numerical data used in our analyses. Unstandardised values offer a more granular understanding of the raw data, allowing readers to comprehend the nuances and intricacies of our study.

Comment:

Please add relative contribution of macronutrient calories (%kcal) to table 2.

Response:

Included a column on the % contribution to total energy to Table 2.

Comment:

Please add data for magnesium, copper, chrome, molybdene, manganese, Vitamin K, biotin.

Response:

We have now included data for magnesium, copper, manganese and Vitamin K1 in Table 2. However, do not have data for chrome, molybdene or biotin.

Comment:

Please add data for uric acid/purines, phytate and other undesired natural food components.

Response:

This is an interesting point to consider. Although we do not have data for these food compounds, we have included a brief sentence in the discussion, highlighting the potential high amounts of phytates in the Santal diet.

Comment:

Please give an estimation of the PAL for people in the respective regions. High intake of digestible carbohydrates is acceptable, where physical activity allows for compensation.

Response:

We have addressed this in the discussion section, where we have highlighted the % of carbohydrates towards the total energy intake (~60%), and we have included references for amount of moderate and heavy physical activity of the Santhal community.

Comment:

Table 3: The table head "macronutrient intake" seems misleading; as the EAT recommendations are certainly not about total macronutrients of specific food groups rather than carbs/fat/protein separately or total amount of specific food groups, this should be amended.

Response:

This was the heading used by the EAT-Lancet paper 'Food in the Anthropocene: the EAT–Lancet Commission on healthy diets from sustainable food systems'. So, to ensure alignment with this paper, we have chosen to keep this heading the same.

Comment:

Discussion: Both patterns lack iodine, B12; maybe more (see missing micronutrients listed above). Khanu Thali provides excessive amounts of salt. Excessive amounts of carbohydrates may be detrimental for patients with risk for / diagnosis of type 2 diabetes or obesity. Please address these topics.

Response:

Thank you for your comment on the dietary patterns, we used the Recommended Dietary Allowances (RDA) for Indians, as indicated in ‘Nutritive Value of Indian Food’ published by NIN in Hyderabad as a reference. We found the menu template were low in iodine, vitamin D and vitamin K1. However, met requirements for the other nutrients, including vitamin B12 (average of 1.6 ug and reccommendations of 1ug). We have discussed these in more detail in the discussion section. The relatively high levels of sodium was addressed in the discussion, as paired with high levels of physical activity they may require higher amounts of sodium due to extra sweat sodium loss. Additionally, we found that the menu templates met requirements for carbohydrate as a % of total energy (60%).

Comment:

Is there epidemiological data about the frequency and magnitude of malnourishment or specific nutrient deficiency in people traditionally following these food patterns?

Response:

In the discussion section, we have included a section on the specific micronutrients deficiencies which are found in the Santhal community, namely iodine, calcium, iron, vitamin B2, folate, and vitamin B12.

Round 2

Reviewer 3 Report

Comments and Suggestions for Authors

The authors have answered to all of the reviewer's suggestions, but revised the manuscript in only a few points.

Few points still remain to be changed in the manuscript as answered in the authors' reply:

Please add relative contribution of macronutrient calories (%kcal) to table 2 for each of the two food schemes, as lines, not columns. Thus, it will be more clear to the readers, that Jhano Thali provides very low amounts of protein (~10 kcal%). This should also be discussed in the final section.

I do not see your added sentence on potentially high phytate content in the discussion.

I do not see your added sentences on potentially low content of iodine, Vitamine K1 and D in the discussion.

I also do not see your extended review of nutrient deficiencies in epidemiological data from Santhal communities.

Comments on the Quality of English Language

minor changes needed

Author Response

Our sincere apologies. Due to some formatting error with track changes, the corrections we had made in the discussion section were not showing up. We have now checked the manuscript and removed track changes, but have highlighted the modified text in yellow. The response to your second round of comments are in green at the end of the attached document. Thanks very much.

Reviewer 3 (Round 2):

Comment:

Please add relative contribution of macronutrient calories (%kcal) to table 2 for each of the two food schemes, as lines, not columns. Thus, it will be more clear to the readers, that Jhano Thali provides very low amounts of protein (~10 kcal%). This should also be discussed in the final section.

Response:

We have included the relative contribution of macronutrients calories (%) as a separate row in table 2, it should now be clear to the reader the proportions of energy of each macronutrient towards each menu template and average.

Comment:
I do not see your added sentence on potentially high phytate content in the discussion.

Response:

Apologies, as there were some missing texts from the discussion. You can now find this addressed on line 317.

Comment:
I do not see your added sentences on potentially low content of iodine, Vitamine K1 and D in the discussion.

Response:

Again apologies, as there were some missing texts from the discussion. You can now find this addressed between lines 361 and 376.

Comment:
I also do not see your extended review of nutrient deficiencies in epidemiological data from Santhal communities.

Response:

You can now find this addressed between lines 355 and 357.